# Study on Multi-Heterogeneous Sensor Data Fusion Method Based on Millimeter-Wave Radar and Camera

**DOI:** 10.3390/s23136044

**Published:** 2023-06-29

**Authors:** Jianyu Duan

**Affiliations:** School of Transportation Science and Engineering, Beihang University, Beijing 100191, China; duanjianyu@buaa.edu.cn

**Keywords:** autonomous vehicle, sensor fusion, uncertainty, perception sensors, camera, radar

## Abstract

This study presents a novel multimodal heterogeneous perception cross-fusion framework for intelligent vehicles that combines data from millimeter-wave radar and camera to enhance target tracking accuracy and handle system uncertainties. The framework employs a multimodal interaction strategy to predict target motion more accurately and an improved joint probability data association method to match measurement data with targets. An adaptive root-mean-square cubature Kalman filter is used to estimate the statistical characteristics of noise under complex traffic scenarios with varying process and measurement noise. Experiments conducted on a real vehicle platform demonstrate that the proposed framework improves reliability and robustness in challenging environments. It overcomes the challenges of insufficient data fusion utilization, frequent leakage, and misjudgment of dangerous obstructions around vehicles, and inaccurate prediction of collision risks. The proposed framework has the potential to advance the state of the art in target tracking and perception for intelligent vehicles.

## 1. Introduction

Multi-source information fusion is a useful technique for processing and integrating data information from multiple information source components [1,2]. It involves techniques such as data detection, target recognition, comprehensive optimization, data association, and tracking processing to effectively coordinate and optimize the data information from different sensors. This allows for the integration of the local information collected by different sensors, while reducing the differences and minimizing redundant information between these data sources [3]. Ultimately, this approach reduces uncertainty and improves the reliability and robustness of the system.

For object recognition and tracking, multi-source information fusion encompasses a range of processes such as raw data acquisition, information interconnectivity, data association, state estimation, and information fusion. To optimize the use of each sensor’s sensing characteristics and exploit the advantages of different principle sensors while mitigating the possibility of system safety issues resulting from common cause failures, an approach relying on heterogeneous sensors is often seen as desirable for information fusion [4].

For intelligent collision avoidance systems, relying solely on the single sensor is limited in ensuring accurate and robust perception in complex road traffic environments [5,6]. Visual perception can capture information about surrounding environment targets based on texture information from images, but it is susceptible to weather conditions such as rain, snow, and backlighting [7]. Millimeter-wave radar can provide precise target speed and distance information and is less susceptible to adverse weather conditions [8]. However, it still has limitations such as low resolution and an inability to classify targets accurately. To address these limitations, multi-source information fusion theory is applied to integrate the advantages of different types of sensors [9,10]. By acquiring information from multiple sources comprehensively and at different levels, collision avoidance systems can achieve high perception accuracy and reliability. The main advantages of information fusion can be summarized as follows.

To improve the reliability of the system and the robustness of the perception module, it is necessary to consider the limitations of perception sensors in complex and dynamic road traffic environments. In such conditions, certain sources of perception may be unusable or have high levels of uncertainty, or they may be outside the sensing range of a particular sensor while another perception sensor can still provide useful information. Information fusion can mitigate these challenges and enable the system to continue working without interruptions, further enhancing the reliability of the perception module.To improve measurement accuracy and enhance target detection capabilities, it is essential to utilize the complementary characteristics of different sensors in a fused system. This approach can effectively improve the accuracy of target recognition and measurement precision.To increase perception reliability and reduce uncertainty, it is necessary to use joint perception information from multiple sensors. This approach can enhance the credibility of detecting targets, increase the redundancy of the perception component, expand the perception coverage, and improve the spatial resolution of perception.For intelligent collision avoidance systems, the key to improving system reliability is how to fully utilize and explore multi-source heterogeneous perception data to reduce the uncertainty of perception and cognition in complex environments.

In this study, we focus on object-level multi-sensor fusion, and the purposed method is based on understanding the spatial synchronization of radar and camera sensor fusion. In particular, we focus on collision avoidance system development using radar–camera sensor fusion. The paper is organized as follows: Section 1 provides an introduction and motivation of the study. Section 2 presents a comprehensive literature review, where various research papers are analyzed and compared. Then, the difficulty and challenge in sensor fusion are summarized. Section 3 discusses the proposed sensor fusion framework in this study. Section 4 discusses the proposed methods for sensor fusion, which includes the object detection, tracking, and fusion algorithms. This part covers the theoretical aspect of implementing the multi-sensor fusion. In Section 5, the experiments are conducted in different scenarios to verify the test and experiment results are analyzed. Finally, the conclusion is given in Section 6.

## 2. Background and Previous Work

Vehicle sensing plays a crucial role in the development of advanced driver-assistance systems (ADASs). The method and performance of varied sensor fusion differs significantly and the architecture decides the perception performance [11]. In reality, most vehicle manufacturers prefer utilizing high-level data fusion architecture for implementing ADAS algorithms in vehicles [12,13]. It is evident that the adoption of appropriate sensor fusion strategies and technologies is essential for ensuring optimal ADAS performance in autonomous vehicles. Sensor fusion plays a vital role in automotive applications. The fusion algorithm’s architecture, methodology, and sensor types depend on the specific task and system requirements. Camera, lidar, ultrasonic sensors, and radar are commonly used sensors to perceive the vehicle’s environment [14]. A powerful and efficient method is purposed, which fuses information from a point cloud generated by LiDAR and image generated by a camera [15]. Haberjahn [16] presented a comprehensive analysis of object-level and low-level sensor fusion, where object-level fusion yields several advantages, such as requiring less computational power and being less sensitive to noise. Thus, object-level sensor fusion is more suitable for real-time embedded systems. Sengupta [17] and Shin [18] demonstrated the benefits of using multiple sensors and fusing their data on the object level. Du [19] conducted research on combining data from various sensors that required temporal and spatial synchronization. They used a geometrical model for spatial synchronization and developed a resolution-matching algorithm based on Gaussian process regression to estimate missing or unreliable data. The sensor fusion process mainly includes the information match, filtering process, and object tracking.

Lots of state-of-the-art methods have been proposed for vehicle sensor fusion [20]. The research conducted by Morris indicates that the utilization of MMW sensors in detecting micro-Doppler signatures is a versatile approach, which effectively distinguishes unmanned aerial systems (UASs) from other moving airborne objects, including birds and other clutter [21]. Cai uses the machine learning method to classify the target for MMW radar, which clearly reveals the trade-off between classification performance and system complexity [22]. García introduces a fail-aware LiDAR odometry system, which has the capability of triggering a safe stop maneuver without requiring driver intervention and can reduce the risk when the system fails [23]. Ren proposes a new lightweight convolution module to improve the vision detection capability [24]. Though the detection performance based on a single sensor has been improved in recent years, the single sensor cannot cover all scenarios due to sensor weakness. Taking advantage of the complementary characteristics of different sensors is necessary to improve perception module reliability. One such method considers the exchange data through vehicle to infrastructure (V2I) communications; Deep Q-Network (DQN) was introduced to predict the optimal minimum contention window in uncertain settings [25]. However, the information fusion process is deployed in the infrastructure, which need not consider the computational resources and burden. Li purposes a sensor-fusion-based vehicle detection and tracking framework at a traffic intersection, which uses the Kalman Filter to fuse the different source data [26]. To track multiple targets using multiple sensors, reference [27] proposes a method combining fuzzy adaptive fusion with wavelet analysis to simplify the linear process model into subsystems, which are estimated separately with multiple KFs. Reference [28] combines the KF with the adaptive neuro-fuzzy inference system to create an accurate target tracking information fusion method, outperforming traditional KF algorithms. In reference [29], predicted states are weighted and averaged, with lower weights given to higher measurement uncertainties. However, the KF is limited to accurately estimating linear systems only and is not suitable for nonlinear systems. A multi-sensor fusion tracking algorithm based on the square root cubature Kalman filter (SRCKF) is purposed for nonlinearity of the vehicle target tracking system [30]. In fact, the motion of traffic participants varies in different scenarios; the motion process model match and noise statistics estimation are crucial for accurate state estimation.

Data association is another key step for the sensor fusion process. Zhao uses the nearest neighbor method in multi-source data fusion to perform data association [31]. The nearest neighbor method cannot address complex scenarios such as occlusion and intersection. In order to create more relationships among sensing data, the maximum likelihood probabilistic data association algorithm is used to achieve multi-object data association with number of targets unknown in advance. Furthermore, Liu purposed a detachable and expansible multi-sensor data fusion model for perception in a level 3 autonomous driving system based on joint probabilistic data association [32]. Due to creating more relations among multi-source data, it is computationally intensive and suitable for more advanced and expensive autonomous systems. However, the current multi-sensor data fusion methods suffer from the high cost of computation resources, low expansibility for more diverse sensors, and insufficient systematic consideration for process modeling.

Overall, multimodal information fusion needs to coordinate the different information data from the different sensor measurements for data fusion. During the filtering process, it is important to establish motion models that match the target’s movement as closely as possible. Data association needs to take into account the uncertainty of heterogeneous sensory data for effective information matching. For target state estimation, it is necessary to consider the statistical characteristics of system noise to improve tracking accuracy, while minimizing algorithm complexity and ensuring real-time computation.

In this study, a novel sensor fusion algorithm is proposed based on radar and camera. This presents a novel data match and target management approach that can consider the target existence uncertainty. To improve the measurement accuracy, the optimization method is introduced to reduce the fusion target state error. To make the motion model more closely match real target motion processes, the improved adaptive interactive multimode cubature Kalman filter is purposed, which can adjust the motion model probability dynamically. To reduce the computational cost and improve real time, the adaptive gate based on vehicle motion is purposed to reduce unnecessary data association. Additionally, tests are conducted in different scenarios for the quantitative assessment of the performance of the proposed sensor fusion algorithm.

## 3. Data Fusion Framework

To ensure the safety of autonomous vehicles, it is fundamental to perform stable and accurate tracking of the surrounding traffic participants, which serves as the basis for subsequent decision-making, trajectory planning, and control operations. For this purpose, a robust tracking system is required that can accurately track targets as soon as they enter the sensor’s perception range, providing accurate information on the target’s position and velocity, and promptly terminate the tracking of invalid targets as soon as they leave the sensor perception range, ensuring the stability and efficiency of the tracking system. The tracking system should also handle issues such as false alarms and missed detections in sensor data that can lead to tracking instability. The proposed multimodal heterogeneous perception cross-fusion framework is designed to improve the obstacle perception and cognition system by data fusion algorithms, tracking filters, and improved data association.

As shown in Figure 1, the information fusion and target tracking process of millimeter-wave radar and camera are presented. After obtaining target perception data from heterogeneous sensors, the data is first time-aligned and the spatial coordinates are transformed. Due to the different spatial coordinate systems and sampling period of millimeter-wave radar and visual sensors, the collected data is not in the same temporal and spatial domain, and thus needs to be aligned in both time and space. After aligning the data from camera and radar, the measurement data from the radar and camera are associated using the Hungarian algorithm based on Euclidean distance. After the visual sensor and radar measurements are matched and fused, the final output information is fused according to the covariance of the measurement data. Meanwhile, the independent target data that is not matched by the millimeter-wave radar and camera are also preserved. In order to reduce false positive and false negative results in the perception fusion, different confidence levels of measurement models are established for different types of data association results, providing uncertain input information for subsequent target tracking fusion methods.

During the target tracking process, the returned detections and established track are associated through the fused information to get more accurate target information. The primary objective of information fusion in track is to reduce uncertainty. The Joint Probabilistic Data Association (JPDA) method is extensively used to exploit the uncertainty obtained from data association and associate multimodal fusion measurement data with the existing confirmed track. In the track management process, the existence probability of the track is calculated using the integrated JPDA method.

## 4. Interacting Multiple Model Adaptive Cubature Kalman Filter

The filtering process is the important step for data fusion and object tracking. In the framework of Bayesian theory, the posterior estimation of a system is iteratively calculated based on the actual measurement likelihood and prior probability estimation. In practical applications, nonlinearity is commonly observed in the system and the statistical properties of noise are often uncertain. The cubature Kalman filter (CKF) algorithm is a well-known method for nonlinear filtering problems, but the algorithm’s covariance matrix may lose positive definiteness during the iteration process, leading to a divergence issue. The key of the CKF is the spherical–radial cubature rule, which makes it possible to numerically compute multivariate moment integrals encountered in the nonlinear Bayesian filter. To address this problem, Arasaratnam [33] proposed the square root cubature Kalman filter (SRCKF) algorithm based on the standard CKF algorithm. The SRCKF algorithm uses the square root factorization of the covariance matrix to obtain a positive-definite matrix, avoiding the potential of non-positive-definite matrices to cause convergence issues during the iteration process. Due to the randomness and mobility of the actual moving target in the complex scenario, the traditional CKF cannot consider the complex object motion process and noise statistics. For improving the object state estimation accuracy, the adaptive model probability and noise characteristics estimation methods are introduced to improve the filtering process.

### 4.1. Adaptive Cubature Kalman Filter

Considering the following nonlinear systems with additive Gaussian noise
(1)xk=f(xk−1)+vk−1
(2)zk=hxk+wk
where xk∈ℝn describes the state vector and zk∈ℝm describes the measurement vector. fx and  hx are known nonlinear system state transition and measurement functions; vk−1 andwk are process and measurement noises, respectively, which are assumed as the uncorrelated zero-mean Gaussian white noises with Qk−1 and Rk covariance.

According to Bayes’ theorem, the posterior distribution of a system state provides complete state statistical information, involving two main steps: time update and measurement update. Firstly, the system state is predicted based on the system model, followed by updating the posterior distribution of the system state with newly obtained measurement information at time *k*. The procedure of the CKF for the nonlinear system can be summarized as:

Time Update(1)Assume that ${\hat{x}}_{k-1}$ and $P_{k-1}$ are known. By Cholesky decomposition, $P_{k-1}$ is decomposed as

(3)$$p(x_{k-1}|D_{k-1})=N({\hat{x}}_{k-1|k-1},P_{k-1|k-1})$$(4)$$P_{k-1|k-1}=S_{k-1|k-1}S_{k-1|k-1}^{T}$$
where ${\hat{x}}_{k-1}$ represents the estimated state at previous time *k*−1 and $P_{k-1}$ represents the system covariance.(2)Estimate the cubature points
(5)$$X_{i,k-1|k-1}=S_{k-1|k-1}\xi_{i}+{\hat{x}}_{k-1|k-1}$$where it entails a total of 2*n* cubature points set (*ξ_i_*, *ω_i_*).(3)Estimate the propagated cubature points and the predicted state
(6)$$X_{{}_{i,k-1|k-1}}^{*}=f(X_{{}_{i,k-1|k-1}},u_{k-1})$$
(7)$${\hat{x}}_{k|k-1}=\frac{1}{m}\sum_{i=1}^{m}X_{i,k|k-1}^{*}$$
(4)Estimate the posterior state covariance
(8)$$P_{k|k-1}=\frac{1}{m}\sum_{i=1}^{m}X_{i,k|k-1}^{*}X_{i,k|k-1}^{*T}-{\hat{x}}_{k|k-1}{\hat{x}}_{k|k-1}^{T}+Q_{k-1}$$

Measurement Update(5)Estimate the cubature points
(9)$$P_{k|k-1}=S_{k|k-1}S_{k|k-1}^{T}$$
(10)$$X_{i,k|k-1}=S_{k|k-1}\xi_{i}+{\hat{x}}_{k|k-1}$$
(6)Estimate the predicted measurement and the square root of the corresponding error covariance
(11)$$Z_{i,k|k-1}=h(X_{i,k|k-1},u_{k})$$
(12)$${\hat{z}}_{k|k-1}=\frac{1}{m}\sum_{i=1}^{m}Z_{i,k|k-1}$$
(7)Estimate the cross-covariance matrix
(13)$$P_{zz,k|k-1}=\frac{1}{m}\sum_{i=1}^{m}Z_{i,k|k-1}Z_{i,k|k-1}^{T}-{\hat{z}}_{k|k-1}{\hat{z}}_{k|k-1}^{T}+R_{k}$$
(14)$$P_{xz,k|k-1}=\frac{1}{m}\sum_{i=1}^{m}X_{i,k|k-1}Z_{i,k|k-1}^{T}-x_{k|k-1}{\hat{z}}_{k|k-1}^{T}+R_{k}$$
(8)Estimate the Kalman gain
(15)$$K_{k}=P_{xz,k|k-1}P_{zz,k|k-1}^{-1}$$
(9)Estimate the updated state and the square root of the corresponding error covariance
(16)$${\hat{x}}_{k|k}={\hat{x}}_{k|k-1}+W_{k}(z_{k}-{\hat{z}}_{k|k-1})$$
(17)$$P_{k|k}=P_{k|k-1}-W_{k}P_{zz,k|k-1}W_{k}^{T}$$

### 4.2. The Noise Statistics Estimation

In the derivation of the root-mean-square algorithm, the statistical characteristics of the process noise and measurement noise are assumed as known constants. However, in real-world scenarios, the noise characteristics are subject to time-varying uncertainties and these uncertainties can significantly impact posterior state estimation. To address this challenge, this study proposes a method that combines maximum likelihood estimation and maximum expectation to estimate noise characteristics. To ensure real-time performance, rolling-time domain estimation is employed to reduce computational complexity. Specifically, an adaptive cubature Kalman filter is proposed to estimate the unknown parameters of the process and measurement noise online. This approach can effectively improve the accuracy of the system state estimation, making it particularly useful in complex and dynamic applications where noise characteristics are highly variable.

As depicted in Figure 2, the system operates based on the initial state *x_k_*_−1_, *P_k_*_−1_, and the noise statistical characteristics *θ_k_*_−1_ = {*R_k_*_−1_, *Q_k_*_−1_}. At time *k*+1, *x_k_* and *P*_k_ are iteratively computed using time and measurement updates, while the unknown noise characteristics are estimated in a sliding time domain using maximum likelihood estimation and maximum expectation methods. Assuming that *θ_k_* = {*R_k_*, *Q_k_*} indicates the estimated statistical characteristics of the process and measurement noises, the parameters can be estimated based on the maximum likelihood criterion formula.
(18)$${\hat{\theta }}_{ML}=\arg\;\max\;L\left(\theta |z_{1:k},x_{1:k}\right)$$
where $L\left(\theta |z_{1:k},x_{1:k}\right)$ is the likelihood function of parameter *θ*.

However, since the system state *x_k_* is unknown, it is not possible to directly solve and calculate the likelihood function. Therefore, the maximum expectation method is used to estimate it. In addition, in order to reduce the computational complexity, the iterative process adopts rolling time domain for iterative computation.

For a fixed time domain *N* > 1, $X_{N}=\left\{x_{j}:j=k-N+1,\cdots ,k\right\}$, $Z_{N}=\left\{x_{j}:j=k-N+1,\cdots ,k\right\}$ represent the system and measurement state. The parameter estimation can be expressed by the following formula:(19)$${\hat{\theta }}_{ML}=\arg\;\max\;L\left(\theta |Z_{N},X_{N}\right)$$

System state transition can be viewed as a first-order Markov process and the likelihood function can be expressed as follows:(20)$$L(\theta |Z_{N},X_{N})=p\left(Z_{N},X_{N}|\theta \right)=p\left(X_{N}|\theta \right)P\left(Z_{N}|X_{N},\theta \right)$$

The maximum expectation (Max-Expect) method mainly includes two steps: expectation calculation and maximum likelihood estimation.

(1)Expectation calculation

Based on the likelihood ratio function and the conditional probability characteristics, it can be described as follows.
(21)$$p\left(Z_{N},X_{N}|\theta \right)=p\left(x_{k-N}|\theta \right)\times \prod_{j=k-N+1}^{k}p\left(x_{j}|x_{j-1},\theta \right)\times \prod_{j=k-N+1}^{k}p\left(z_{j}|x_{j},\theta \right)$$

During the iteration process, $p\left(x_{k-N}|\theta \right)$ is the initial probability distribution of the system state in the time domain and the initial state of the system follows a Gaussian distribution with $x_{k-N}\sim N\left({\hat{x}}_{k-N},\;P_{k-N}\right)$.
(22)$$P\left(x_{k-N}|\theta \right)={\left(2\pi \right)}^{-\frac{n}{2}}{\left|P_{k-N}\right|}^{-\frac{1}{2}}\times \exp\left\{-\frac{{\left\|x_{k-N}-{\hat{x}}_{k-N}\right\|}_{P_{k-N}^{-1}}^{2}}{2}\right\}$$
where there is a total of 2n cubature points. The square root Kalman filter has a Gaussian filtering property and the probability $p\left(x_{j}|x_{j-1},\theta \right)$ can be calculated by the state prediction equation. The logarithmic likelihood function is shown as follows.
(23)$$\begin{array}{ll} \ln\left[L\left(\theta |Z_{N},X_{N}\right)\right] & =C-\frac{k}{2}\ln\left|Q\right|\\ & -\frac{1}{2}\sum_{j=k-N+1}^{k}{\left\|x_{j}-f\left(x_{j-1}\right)\right\|}_{Q^{-1}}^{2}-\frac{N}{2}\ln\left|R\right|\\ & -\frac{1}{2}\sum_{j=k-N+1}^{k}{\left\|z_{j}-h\left(x_{j-1}\right)\right\|}_{R^{-1}}^{2} \end{array}$$
where, $\;C=-\frac{k\left(n+m\right)+m}{2}\ln\left(2\pi \right)-\frac{1}{2}\ln\left|P_{k-N}\right|-\frac{\|x\left(k-N\right)-\hat{x}\;\left(k-N\right)\;{\|}_{P_{k-N}}^{2}}{2}$ is a constant.

Calculating expectation


(24)
$$\begin{array}{ll} J=E\left\{\ln\left[L\left[\theta |Z_{N},X_{N}\right]\right]\right\} & =C-\frac{N}{2}\ln\left|Q\right|-\\ & -\frac{1}{2}\sum_{j=k-N+1}^{k}\left\{E{\left\|x_{j}-f\left(x_{j-1}\right)\right\|}_{Q^{-1}}^{2}\right\}-\frac{N}{2}\ln\left|R\right|\\ & -\frac{1}{2}\sum_{j=k-N+1}^{k}\left\{E{\left\|z_{j}-h\left(x_{j}\right)\right\|}_{R^{-1}}^{2}\right\} \end{array}$$


(2)Maximum likelihood estimation

This step aims to estimate parameters by solving for the maximum value of the logarithmic likelihood function.
$$\frac{\partial J}{\partial Q}=0$$
(25)$$\frac{\partial J}{\partial R}=0$$

Furthermore
(26)$${\hat{Q}}_{k}=\frac{1}{N}\sum_{j=k-N+1}^{k}\left\{\begin{array}{cc} \left[x_{j}-f\left(x_{j-1}\right)\right] & {\left[x_{j}-f\left(x_{j-1}\right)\right]}^{T} \end{array}\right\}$$
(27)$${\hat{R}}_{k}=\frac{1}{N}\sum_{j=k-N+1}^{k}\left\{\begin{array}{cc} \left[z_{j}-h\left(x_{j-1}\right)\right] & {\left[z_{j}-h\left(x_{j-1}\right)\right]}^{T} \end{array}\right\}$$
where $f\left(x_{j-1}\right)$ represents the computation of the state transition value at the cubature point $x_{j-1}$.

Traditional maximum likelihood algorithms also require smoothing filtering in the rolling time domain. In order to further reduce computational effort, the estimation of system noise characteristics is directly replaced by the estimated value at time *k*. Thus, the estimated system noise characteristics are as follows.
(28)$$\begin{array}{cc} {\hat{Q}}_{k/N} & =\frac{1}{N}\left\{\left(N-1\right){\hat{Q}}_{k-1}\right.+diag\left\{\left({\hat{x}}_{k}{\hat{x}}_{k}^{T}+P_{k}\right)\right.\\ & -\left[\frac{1}{2n}\sum_{i=1}^{2n}f\left(X_{i,k-1}\right)\times X_{i,k-1}^{T}\right]\\ & -\left[\frac{1}{2n}\sum_{i=1}^{2n}X_{i,k-1}^{T}\times f\left(X_{i,k-1}\right)\right]\\ & +\left[\frac{1}{2n}\sum_{i=1}^{2n}f\left(X_{i,k-1}\right)\times f{\left(X_{i,k-1}\right)}^{T}\right]\left.\left.\right\}\right\} \end{array}$$
(29)$$\begin{array}{cc} {\hat{R}}_{k/N} & =\frac{1}{N}\left\{\left(N-1\right){\hat{R}}_{k-1}\right.+diag\left\{{\hat{z}}_{k}{\hat{z}}_{k}^{T}\right.\\ & -\left[\frac{1}{2n}\sum_{i=1}^{2n}h\left(X_{i,k}\right)\times z_{k}^{T}\right]\\ & -\left[\frac{1}{2n}\sum_{i=1}^{2n}z_{k}\times h{\left(X_{i,k-1}\right)}^{T}\right]\\ & +\left[\frac{1}{2n}\sum_{i=1}^{2n}h\left(X_{i,k}\right)\times h{\left(X_{i,k}\right)}^{T}\right]\left.\left.\right\}\right\} \end{array}$$

### 4.3. Adaptive Interactive Multiple Model Motion Prediction

During the time update process of the Kalman filter, it is necessary to predict the target motion state. For traffic objects, constant velocity, constant acceleration, and constant speed motion are common motion models.

The steady-state motion model represents the target moving at a constant speed along a straight line, considering only planar motion for simplification of calculation. The discrete motion equation for this model is shown in the formula.
(30)$$\left[\begin{array}{c} x_{k+1}\\ v_{x,k+1}\\ y_{k+1}\\ v_{y,k+1} \end{array}\right]=\left[\begin{array}{cccc} 1 & T & 0 & 0\\ 0 & 1 & 0 & 0\\ 0 & 0 & 1 & T\\ 0 & 0 & 0 & 1 \end{array}\right]\left[\begin{array}{c} x_{k}\\ vx_{k}\\ y_{k}\\ vy_{k} \end{array}\right]+\left[\begin{array}{cc} \frac{1}{2}T^{2} & 0\\ T & 0\\ 0 & \frac{1}{2}T^{2}\\ 0 & T \end{array}\right]v(k)$$

The steady-state acceleration motion model represents the target moving along a straight line with uniform acceleration. The discrete motion equation for this model is shown as follows.
(31)$$\left[\begin{array}{c} x_{k+1}\\ vx_{k+1}\\ ax_{k+1}\\ y_{k+1}\\ vy_{k+1}\\ ay_{k+1} \end{array}\right]=\left[0\begin{array}{cccccc} 1 & T & \frac{1}{2}T^{2} & 0 & 0 & 0\\ 0 & 1 & T & 0 & 0 & 0\\ 0 & 0 & 1 & 0 & 0 & 0\\ 0 & 0 & 0 & 1 & T & \frac{1}{2}T^{2}\\ 0 & 0 & 0 & 0 & 1 & T\\ 0 & 0 & 0 & 0 & 0 & 1 \end{array}\right]\left[\begin{array}{c} x_{k}\\ vx_{k}\\ ax_{k}\\ y_{k}\\ vy_{k}\\ ay_{k} \end{array}\right]+\left[\begin{array}{cc} \frac{T^{3}}{6} & 0\\ \frac{T^{2}}{2} & 0\\ T & 0\\ 0 & \frac{T^{3}}{6}\\ 0 & \frac{T^{2}}{2}\\ 0 & T \end{array}\right]v(k)$$

The steady-state uniform circular motion represents the target moving at a constant speed in a circular path and the discrete motion equation for this model is shown as follows.
(32)$$\left[\begin{array}{c} x_{k+1}\\ v_{x,k+1}\\ y_{k+1}\\ v_{y,k+1} \end{array}\right]=\left[\begin{array}{cccc} 1 & \frac{\sin\omega T}{\omega } & 0 & \frac{\cos\omega T-1}{\omega }\\ 0 & \cos\omega T & 0 & -\sin\omega T\\ 0 & \frac{1-\cos\omega T}{\omega } & 1 & \frac{\sin\omega T}{\omega }\\ 0 & \sin\omega T & 0 & \cos\omega T \end{array}\right]\left[\begin{array}{c} x_{k}\\ vx_{k}\\ y_{k}\\ vy_{k} \end{array}\right]+\left[\begin{array}{cc} \frac{T^{2}}{2} & 0\\ T & 0\\ 0 & \frac{T^{2}}{2}\\ 0 & T \end{array}\right]w_{k}$$

The above three models have corresponding physical processes and, when the actual motion of the target matches the process model, it can predict the motion state well. However, due to the complexity and randomness of target motion, a single model cannot accurately update the target state over time. Unlike single-model algorithms, multiple-model interaction algorithms assume that the target motion process is composed of several motion models at each moment. When the target motion state changes in real time, the multiple-model interaction algorithm adjusts the model probability through a Markov chain to adapt to the current target motion state.

As shown in Figure 3, the flowchart of the multi-model interaction algorithm is presented. Firstly, the model interaction and conditional initialization calculation are performed. Predictions and filtering are carried out for each motion model. The model probability estimator updates the probability of the motion model. Finally, the weighted calculation is performed on the state estimates of each model’s filtering to output the final fusion state. The detailed algorithm flow is shown as follows.

Model interaction: the model conditions such as system state and covariance can be obtained from all filters at the previous time *k*−1.(33)$${\hat{X}}_{j}^{0}(k-1|k-1)=\sum_{i=1}^{r}{\hat{X}}^{i}(k-1|k-1)\mu_{i|j}(k-1|k-1)$$(34)$$\begin{array}{c} P_{j}^{0}(k-1|k-1)=\sum_{i=1}^{r}\mu_{i|j}(k-1|k-1)\\ \;\;\;\;\;\;\;\;\;\;\;\;\;\;\left\{P^{j}(k-1|k-1)+[{\hat{X}}^{i}(k-1|k-1)-{\hat{X}}_{j}^{0}(k-1|k-1)]\right\}\\ \;\;\;\;\;\;\;\;\;\;\;\;\;\;\left.{\left[{\hat{X}}^{i}(k-1|k-1)-{\hat{X}}_{j}^{0}(k-1|k-1)\right]}^{T}\right\} \end{array}$$where ${\hat{X}}_{j}^{0}\left(k-1|k-1\right)$ represents the integrated estimated state of model *j* at time *k*−1, $P_{j}^{0}\left(k-1|k-1\right)$ is its initial covariance, and $\mu_{i|j}\left(k-1\right)$ denotes the transition probability from model *i* to model *j* at time *k*−1.
(35)$$\mu_{i|j}(k-1|k-1)=\frac{1}{{\bar{c}}_{j}}p_{ij}\mu_{i}(k-1)$$
(36)$${\bar{c}}_{j}=\sum_{i=1}^{r}p_{ij}\mu_{i}(k-1)$$

The transition between motion models follows a first-order Markov chain; $\mu_{i}\left(k-1\right)$ represents the probability value matched with model *i* at time *k*−1, $p_{ij}$ represents the probability of transitioning from model *i* to model *j*, and the Markov transition probability matrix is defined as follows.
(37)$$P=\left[\begin{array}{ccc} p_{11} & \cdots & p_{1r}\\ \vdots & \ddots & \vdots \\ p_{r1} & \cdots & p_{rr} \end{array}\right]$$

2.Model matched prediction update: based on the mixed initial state estimation and measurements, motion prediction state and covariance are calculated for each motion model.


(38)
$${\hat{X}}^{j}(k|k-1)=\Phi^{j}(k-1){\hat{X}}_{j}^{0}(k-1|k-1)$$



(39)
$$P^{j}(k|k-1)=\Phi^{j}(k-1)P_{j}^{0}(k-1|k-1)\Phi^{j}{\left(k-1\right)}^{T}+Q^{j}(k-1)$$



(40)
$${\hat{X}}^{j}(k|k)=X^{j}(k|k-1)+K^{j}(k)v^{j}(k)$$



(41)
$$P^{j}(k|k)=P^{j}(k|k-1)-K^{j}(k)v^{j}(k)$$



(42)
$$P^{j}(k|k)=P^{j}(k|k-1)-K^{j}(k)S^{j}(k)K^{j}{\left(k\right)}^{T}$$


3.Model probability update: the likelihood for each model can be calculated using the error covariance and the mean error. Assuming it follows the Gaussian distribution, then the likelihood function of model j is shown as follows.


(43)
$$\Lambda_{j}(k)=\frac{1}{\sqrt{2\pi }{\left|S^{j}(k)\right|}^{1/2}}\exp\{-\frac{1}{2}v_{j}^{}(k)S^{j}{\left(k\right)}^{-1}v_{j}^{T}(k)\}$$


The likelihood function describes the probability of observing a set of data given a certain set of unknown parameters; $V_{k}^{j}$ represents the mean error and $S_{k}^{j}$ represents the corresponding covariance matrix.

The model probability is updated using the estimation model probability and likelihood function.
(44)$$\mu_{j}(k)=\frac{1}{c}\Lambda_{j}(k){\bar{c}}_{j}$$
(45)$$c=\sum_{j=1}^{r}\Lambda_{j}(k){\bar{c}}_{j}$$

4.Posterior state estimation


(46)
$$\begin{array}{l} \hat{X}(k|k)=\sum_{j=1}^{r}\mu_{j}(k)\{P^{j}(k|k)+[{\hat{X}}^{j}(k|k)-\hat{X}(k|k)]\\ {\left[{\hat{X}}^{j}(k|k)-\hat{X}(k|k)\right]}^{T}\} \end{array}$$


In the multi-mode interaction algorithm, the model switching follows a first-order Markov process. In the process of state transition, the probability matrix of state transition is crucial for system mode selection and switching. Usually, the state transition probability matrix is pre-set based on experience and cannot be updated online in real time. In order to adapt more accurately to the target’s real motion process, the state transition probability matrix is online-corrected in the transition process.
(47)$$\begin{array}{cc} {\hat{p}}_{ji} & =P\{m_{i}(k+1)|m_{j}(k),z(k)\}\\ & =\frac{\Lambda_{ji}(k+1)|P\{(m_{i}(k+1)|m_{j}(k),z^{k})\}}{P(z(k+1)|m_{j}(k),z^{k})} \end{array}$$
(48)$$\Lambda_{ji}(k+1)=\frac{1}{\sqrt{2\pi }{\left|S^{ji}(k+1)\right|}^{1/2}}\exp\{-\frac{1}{2}v_{ji}^{T}(k+1)S^{ji}{\left(k+1\right)}^{-1}v_{ji}^{}(k+1)\}$$

Considering the independence of measurement sampling, $z^{k}$ contains the model matching information before time *k*; thus, the probability can be described as follows.
(49)$$P\{m_{i}(k+1)|m_{j}(k+1),Z^{k}\}=P\{m_{i}(k+1)|m_{j}(k+1)=p_{ji}$$
(50)$${\hat{p}}_{ji}=p_{ji}\times \Lambda_{ji}(k+1)/\beta_{j}(k)$$
(51)$$\beta_{j}(k)=\sum_{i}\Lambda_{ji}(k+1)\times p_{ji}$$

Taking the tracking process of the target vehicle’s lane-changing as an example for simulation analysis, the vehicle initially moves at a constant speed in a straight line, then changes lane using a fifth-order polynomial curve and, after the lane change, it decelerates. The simulation measurement sensor is located at the coordinate origin point and the sampling time for the measurement data is 10 ms. The distance measurement error is 0.2 m and the angle measurement error is 0.1 rad. The proposed AIMM-ASRCKF algorithm and the IMM-CKF algorithm are both subjected to 100 Monte Carlo simulations. Figure 4 shows a comparison between the true trajectory and the filtered trajectory. It can be observed that both the IMM-CKF and AIMM-ASRCKF have good filtering effects when the target vehicle moves in a straight line. However, when the target vehicle is starting or ending the lane change behavior, the IMM-CKF has a larger filtering error, while the proposed AIMM-ASRCKF has a better filtering effect.

Figure 5 and Figure 6 show the tracking accuracy in the X and Y directions. Overall, the purposed AIMM-ASCRCKF has smaller tracking errors compared to the IMM-CKF. It is noteworthy that the AIMM-ASCRCKF has a significantly faster convergence rate than the IMM-CKF at the beginning of the filtering, which is mainly due to the proposed algorithm’s strong model adaptability. Additionally, there are significant tracking errors in both filtering methods during the lane-changing process due to the significant change in motion pattern.

To further quantify the filtering effect, the root mean square error (RMSE) is used to measure the effectiveness of different filtering algorithms. Table 1 shows the results and its calculation process is shown in the formula.
(52)$$RSME=\sqrt{\frac{1}{M}\sum_{k=1}^{M}{\left(\hat{x}-x\right)}^{2}}$$
where *M* is the number of Monte Carlo simulations, $\hat{x}$ represents the estimated state, and *x* is the true value.

Figure 7 show the model probabilities of the two algorithms. Since the adaptive multiple model interaction algorithm can calculate the state transition probability online, it can quickly switch models when the target motion mode changes, making the model closer to the true physical process of motion, and the algorithm has strong model adaptability.

### 4.4. Improved JPDA Algorithm considering Uncertainty Fusion

The key to multi-object tracking algorithms is associating measurement information with targets. The JPDA algorithm achieves data association by computing the probability of each measurement–target association event. However, with increasing numbers of targets, the traditional JPDA algorithm exhibits exponential growth in association events, leading to the combinatorial explosion phenomenon. Furthermore, the JPDA algorithm assumes a constant value for target detection probability. In practical complex environments, the target detection probability varies with changes in the external environment. Due to the characteristics of sensors, real targets may disappear temporarily or false targets may appear intermittently, making it difficult for the traditional JPDA algorithm to manage target tracks dynamically. To address the aforementioned issues, this study proposes an improved JPDA algorithm suitable for multi-sensor information perception. The purposed algorithm corrects the probability of association events based on the confidence of measurement information matching results. Additionally, an adaptive gating is applied to determine effective association events and posterior estimation is carried out through the establishment of a unified uncertain information measurement model.

The success of multi-object tracking algorithms relies heavily on the association of measured data with targets. The JPDA algorithm achieves data association by computing the probability of each measurement–target association event. However, as the number of targets grows, the traditional JPDA algorithm results in exponential increases in association events, leading to a combinatorial explosion phenomenon. Furthermore, the JPDA algorithm assumes a constant value for target detection probability. In real complex environments, the target detection probability varies with changes in the external environment. Due to the characteristics of sensors, real targets may disappear temporarily or false targets may appear intermittently, making it difficult for the traditional JPDA algorithm to manage target trajectories dynamically.

The flowchart of the proposed improved JPDA algorithm is shown in Figure 8. Similar to the traditional JPDA algorithm, the algorithm mainly includes prediction, association, and update. The prediction and update parts are the same as those in traditional JPDA, and are calculated based on Bayesian theory. The confirmation of data association events and the calculation of conditional probability are the key of the proposed algorithm.

The set of association events can be defined as follows.
(53)$$\Omega =[\omega_{jt}],\;j=1,\;...,\;m,\;t=1,...,T$$
where $\omega_{jt}=1$ indicates that measurement *j* falls within the association gate domain for target *t* and $\omega_{jt}=0$ indicates that measurement *j* does not fall within the association gate domain for target *t*.

When calculating the conditional probability of association events, the following assumptions are made:(1)Each measurement can only originate from a unique true target or is not associated with any existing track.(2)Each target can correspond to at most one measurement. Thus, a large number of association events occur during the process of associating measurements with true targets. As the number of measurements and true targets increases, calculating the conditional probability of association events exponentially grows. The proposed algorithm addresses this issue by using an adaptive gating technique to filter out unlikely association events, thereby ensuring real-time processing.

When calculating the probability of data correlations, it mainly includes the following steps:(1)Create the association event confirmation matrix based on the current measurement information and the key matching pairs of the previous measurement.(2)Calculate the conditional probability of association events, assuming there are *N* targets within the tracking field of view, where the target tracking gate can be established at the predicted positions of *N* targets at time *k*. Among them, m measurement results fall within the target tracking gate field.

The conditional probability of association events can be defined as follows.
(54)$$\beta_{jt}(k)=\left\{\begin{array}{c} \frac{P_{D}P_{G}}{d_{jt}^{2}(k)V^{nt-1}},\omega_{jt}=1,j\neq 0\\ \frac{1-P_{D}P_{G}}{V^{nt}},\omega_{jt}=1,j=0\\ 0,\omega_{jt}=0 \end{array}\right.$$
where $P_{D}$ is the probability of target detection, $P_{G}$ is the probability of the measurement falling within the tracking gate, $V^{nt}$ is the noise clutter statistical model, and $d_{jt}^{2}$(*k*) represents the Mahalanobis distance between the measurement value $z_{j}\left(k\right)$ at time k and the predicted value $\hat{z}(k|k-1)$ of target *t*.
(55)$${\tilde{V}}_{k}(\gamma )≜d_{jt}^{2}(k)={\left[z_{j}(k)-{\hat{z}}_{t}(k|k-1)\right]}^{'}S_{t}^{-1}(k)\times [z_{j}(k)-{\hat{z}}_{t}(k|k-1)]$$

$S_{t}$ represents the new information covariance of the target at time *k* in the Kalman filter process. If the dimension of the measurement state is n_z_, the variable ${\tilde{V}}_{k}\left(\gamma \right)$ follows a $\chi^{2}$ distribution with n_z_ degrees of freedom.

In order to reduce the computational burden of data association, an association gate is usually set to determine whether the measurement information falls within the predicted region of the relevant target. Figure 9 shows a schematic diagram of the association gate.

The measurements that fall within the association gate are considered as valid associated measurements.
(56)$${\tilde{V}}_{k}(\gamma )\leq \gamma$$

The probability *P_G_* can be calculated as follows.
(57)$$P_{G}=\mathrm{Pr}\left\{z_{k}^{j}\in {\tilde{V}}_{k}(\gamma )\right\}$$

The parameter γ directly influences the size and probability of the association gate region and the association gate can be adjusted in real time by dynamically adjusting the value of γ.
(58)$$\hat{\gamma }=v_{k}^{'}S_{k}^{-1}v_{k}$$

Perform Cholesky decomposition on $S_{k}$, $S_{k}=W^{\prime }W$
(59)$$\begin{array}{cc} \hat{\gamma } & =v_{k}^{'}{\left(W^{'}W\right)}^{-1}v_{k}\\ & ={\left[{\left(W^{-1}\right)}^{'}v_{k}\right]}^{'}\left[{\left(W^{-1}\right)}^{'}v_{k}\right] \end{array}$$

Adaptive association gate boundary parameter γ
(60)$$\gamma =\frac{\hat{\gamma }}{\left|W\right|}$$

(3)Correction of associated event probabilities

To address the uncertainty in data association arising from fusion of multisource information, we introduce a correction factor *λ* that characterizes this uncertainty and use it to adjust our association probabilities. We assign the highest confidence level to the fusion measurement results that match both the visual sensor and millimeter-wave radar sources, while visual-only observation is considered at a relatively high confidence level, and millimeter-wave radar-only observation is considered at a lower confidence level. The lowest confidence level is assigned to sets of measurements that cannot be matched. The adjusted correlation event probability is then computed based on these calibration factors. The corrected associate event probability can be defined as follows.
(61)$$\beta_{jt}(k)=\left\{\begin{array}{c} \frac{\lambda P_{D}P_{G}}{d_{jt}^{2}(k)V^{nt-1}},\omega_{jt}=1,j\neq 0\\ \frac{1-\lambda P_{D}P_{G}}{V^{nt}},\omega_{jt}=1,j=0\\ 0,\omega_{jt}=0 \end{array}\right.$$

(4)Normalization processing


(62)
$$\beta_{jt}^{'}(k)=\frac{\beta_{jt}(k)}{\sum_{j=0}^{m}\beta_{jt}(k)}$$


(5)Object state estimation


(63)
$${\hat{X}}^{j}(k|k)=\sum_{i=0}^{m}\beta_{ij}^{'}{\hat{X}}_{i}^{j}(k|k)$$


In order to evaluate the accuracy of tracking algorithms, the GOSPA evaluation metric is introduced. Assuming there are m ground truth objects and n tracks at time *k*, where *m ≤ n*, the GOSPA is defined as follows.
(64)$$\mathrm{GOSPA}={\left[\sum_{i=1}^{m}d_{c}^{p}(x_{i},y_{\pi (i)})+\frac{c^{p}}{\alpha }(n-m)\right]}^{1/p}$$
where *d_c_* represents the truncation distance, *p* represents sensitivity to outliers in the localization component, and $y_{\pi \left(i\right)}$ denotes the assignment of track *i* to the ground truth object *x_i_*. When *α* = 2, GOSPA can be decomposed into state error, missed detection error, and false alarm error. GOSPA can be simplified as:(65)$$\mathrm{GOSPA}={\left[loc^{p}+miss^{p}+false^{p}\right]}^{1/p}$$

Figure 10 shows a comparison of fusion results, which indicates that the proposed fusion algorithm performs better in terms of the GOSPA metric compared to single perception sensors. By further analyzing the localization error, missed detection error, and false detection error, it is found that the single camera sensor has a larger localization error and some degree of missed detection, while the millimeter-wave radar produces false detections due to clutter. The proposed fusion algorithm effectively integrates the advantages of both sensors, achieving good tracking accuracy and precision.

## 5. Experimental Test and Validation

### 5.1. Heterogeneous Sensor Spatio-Temporal Synchronization

In the process of heterogeneous sensor target-level information fusion, it is necessary to send data from different types of sensors to the fusion center. As the detection processes of each sensor are independent and non-interacting, it is necessary to align the asynchronous data in terms of time and unify them in space.

Time alignment refers to synchronizing the measurement information of the same target detected by heterogeneous sensors to the same moment. Since the working principles of heterogeneous sensors are different and their measurement processes are independent of each other, the reporting cycles to the fusion center for target information are different. Before information fusion, it is necessary to align the asynchronous sensor measurement information to the same moment. The cycle period of millimeter-wave radar data is approximately 16 ms, with a relatively short and stable interval. The cycle period of visual sensors is longer and can fluctuate with an increase in the number of targets in the scene. The millimeter-wave radar timestamp is selected as the standard for time alignment and the visual sensor perception information is converted into time series. For visual perception information, the least-squares cubic spline curve is used for fitting the interpolation calculation. That is to say, the data measured n+1 times by the camera output in the time range [a, b] is fitted to obtain the function S(x). Then, the sampling time of the millimeter-wave radar data is used as the independent variable to input into the fitting function to obtain the visual sensor data information corresponding to the corresponding moment. After obtaining the target output information of the millimeter-wave radar and the visual sensor at the same moment, spatial transformation is performed to place the measurement results in the same spatial coordinate system.

Due to the different installation positions of camera and millimeter-wave radars on the vehicle, they detect targets in their respective coordinate systems. Before data fusion, it is necessary to unify the target information detected by each sensor in the same spatial coordinate system. As shown in Figure 11, a unified vehicle coordinate system is defined with the center of the rear axle as the origin, and the target information from the visual sensors and millimeter-wave radars is transformed into the vehicle coordinate system. Since the target data of camera and millimeter-wave radars do not involve Z-axis information perpendicular to the ground, the coordinate transformation process only considers the XY plane. In addition, the camera and millimeter-wave radars are both installed on the vehicle’s centerline, so the coordinate transformation mainly considers translation in the X-axis direction.

### 5.2. Optimal Estimation of Object-Level Information Fusion

After spatio-temporal unification of camera and millimeter-wave radars, the matched target information needs to be fused and estimated. As shown in Figure 12, it is a schematic diagram of information fusion from matched millimeter-wave radar and camera. The depth measurement information of the visual sensor has significant uncertainty, and the millimeter-wave radar has large uncertainty in the measured information of the horizontal direction due to its lower angular resolution. The fusion algorithm can optimize the accuracy to minimize the error based on the sensor uncertainty.

By integrating the measurement characteristics of sensors, the maximum likelihood estimation principle is used to optimize target information. The measurement information of the camera and millimeter-wave radar are defined as *M_C_* and *M_R_*, respectively, and the likelihood function *L*(*M_t_*) is the likelihood function of the estimated quantity x. *P*(*M_C_*) and *P*(*M_R_*) are the distribution functions of the camera and millimeter-wave radar, respectively. Assuming that the sensor measurement data follow a Gaussian distribution, their conditional probability can be expressed as follows.
(66)$$P(M_{C}|M_{t})=\frac{1}{\sqrt{2\pi }\sigma_{1}^{2}}e^{-\frac{{\left(M_{C}-\mu \right)}^{2}}{2\sigma_{1}^{2}}}$$
(67)$$P(M_{R}|M_{t})=\frac{1}{\sqrt{2\pi }\sigma_{2}^{2}}e^{-\frac{{\left(M_{R}-\mu \right)}^{2}}{2\sigma_{2}^{2}}}$$

*M_t_* represents the true value of the target state; $\sigma_{1}$ and $\sigma_{2}$ are the measurement variances of the camera and millimeter-wave radar. In the measurement process, assuming that each sensor is independent of each other, their posterior likelihood function can be expressed as follows.
(68)$$\begin{array}{cc} P(M_{t}) & =P(M_{C}/M_{t})P(M_{R}/M_{t})\\ & =\frac{1}{2\pi \sigma_{1}^{2}\sigma_{2}^{2}}e^{\frac{-{\left(M_{C}-\mu \right)}^{2}}{2\sigma_{1}^{2}}+\frac{-{\left(M_{R}-\mu \right)}^{2}}{2\sigma_{2}^{2}}} \end{array}$$

The logarithmic likelihood function can be expressed as follows.
(69)$$\begin{array}{l} L(M_{t})=\sum_{i=1}^{n}\log p(M_{n}/M_{t})\\ =\sum_{i=1}^{n}-\frac{1}{2}\log[{\left(2\pi \right)}^{\frac{n}{2}}|P_{i}|]\left(-\frac{1}{2}{\left(M_{n}-M_{t}\right)}^{'}P_{i}^{-1}\left(M_{n}-M_{t}\right)\right) \end{array}$$
where *P_i_* is the covariance matrix, *n* is the number of sensors, and *M_n_* is the measured results of sensor *n*.

By solving the logarithmic maximum likelihood function, the estimated state variables can be obtained.
(70)$$X_{mle}=\frac{\sum_{i=1}^{n}P_{i}^{-1}M_{n}}{\sum_{i=1}^{n}P_{i}^{-1}}$$

Since the information fusion in this study only involves two sensors, the millimeter-wave radar and camera, the optimized state estimation can be represented as:(71)$$X_{mle}=P_{C}{\left(P_{C}+P_{R}\right)}^{-1}M_{C}+P_{R}{\left(P_{C}+P_{R}\right)}^{-1}M_{R}$$
where *P_C_* is the covariance of measurements from the visual sensor and *P_R_* is the covariance of measurements from the millimeter-wave radar.

### 5.3. Experiments and Results Analysis

The experimental test vehicle platform is built as shown in Figure 13. The vehicle is equipped with a Continental ARS 408 mm-wave radar, a Mobileye EyeQ3 smart camera, and an 80-line LiDAR, all of which communicate via CAN bus. The millimeter-wave radar is installed at the center of the vehicle’s front bumper, with a height of 180 mm from the ground, while the camera is mounted at the top centerline of the windshield. To ensure accurate measurement accuracy, the positions of the radar and camera need to be calibrated during the installation process. With the vehicle coordinate system as a reference, the sensor installation position and angle are finely adjusted to ensure that the XY plane of the vehicle coordinate system is parallel to the XY plane of the sensor coordinate system. In addition, due to the strong anti-interference ability and high measurement accuracy of the LiDAR, the LiDAR measurement data is used as the ground truth for comparative analysis.

As shown in Figure 14, a Speedgoat real-time target machine was used as the real-time fusion processing platform. The information from the millimeter-wave radar, camera, and vehicle state information were transmitted to the Speedgoat real-time target fusion computing platform via a CAN bus for information fusion. The Vector CAN bus tool is used for online monitoring and data logging.

The heterogeneity of sensor perception characteristics and measurement accuracy may result in differences in detection effectiveness under different scenarios, especially in extremely complex situations where one sensor may temporarily malfunction. Multimodal information fusion of heterogeneous sensors can effectively improve the system’s robustness and measurement accuracy. To demonstrate the effectiveness of the fusion algorithm, the experiment results among radar, camera, and fusion algorithm are compared in complex test scenarios. The testing scenarios include crossroads, pedestrian crossings, abnormal weather, and other situations, while GOSPA evaluation metrics are used to uniformly compare the results.

(1)The crossroad scenario

There are a large number of complex traffic participants in the crossroad and the purposed fusion algorithm is mainly validated in this scenario. As shown in Figure 15, the test results indicate there is a significant improvement in the GOSPA comprehensive score for the fusion system. From the GOSPA-Missed score perspective, it can be seen that the camera is more prone to miss detections compared to the millimeter-wave radar, resulting in a higher GOSPA score overall for the camera. On the other hand, the millimeter-wave radar is more prone to false alarms, which is an important component of its GOSPA score. In addition, from the GOSPA-loc score perspective, the camera has a larger deviation in obtaining target state information, while the fused target information has higher accuracy.

(2)Pedestrian crossing scenario

Traffic accidents are more likely to occur in the pedestrian crossing scenario. The testing purpose is to examine the response speed and measurement accuracy of the fusion algorithm in such extreme scenarios. As shown in Figure 16, in the pedestrian crossing scenario, the camera can perceive pedestrian targets first, while the millimeter-wave radar is relatively delayed in recognizing pedestrians due to its perceptual characteristics. The fused perception system can identify pedestrian targets earlier based on the perception and recognition results of the camera and have the high-resolution characteristics combining the millimeter-wave radar advantage of measurement accuracy, resulting in a lower positioning error.

(3)Nighttime scenario

The primary goal of the test in the dark environment is to evaluate the reliability of the fusion perception algorithm when a single sensor fails. As shown in Figure 17, the test results show that the camera has poor target recognition at night, while the millimeter-wave is relatively stable. The results indicates that the proposed fusion algorithm can still identify and detect road targets when a single sensor fails.

(4)Underground parking scenario

Due to the relatively enclosed nature of the underground parking scenario, there is more reflection of echo information by the millimeter-wave radar, which makes it prone to missed detections and false detections. As depicted in Figure 18, the millimeter-wave radar shows a higher rate of misidentification, while the proposed fusion algorithm can efficiently eliminate clutter targets, thereby reducing the chances of false alarms. Actually, the fusion algorithm still generates a few false alarms due to the unified fusion process; it can be adjusted flexibly based on scenario understanding in a future study.

Based on the comprehensive results of various scenario tests, the proposed fusion algorithm exhibits high detection performance and measurement accuracy in complex and extreme environments. In the crossroad scenario, there is a diverse range of traffic participants and targets can occlude each other, hindering their perception and detection by the perception module. In such testing environments, it is beneficial for evaluating the comprehensive performance of the perception module. The results show that the proposed algorithm can fully utilize the strengths of the heterogeneous sensors and achieve stable target detection and tracking. In the pedestrian crossing scenario, pedestrians can suddenly appear from the blind spots of the perception module. Because the fusion module can use the camera detection result immediately, the relevant detection information can be promptly transmitted to the decision-making and planning module. Also, there is some delay compared with the single camera detection in this test, because the fusion algorithm needs time to process the comprehensive detection results again. In the nighttime scenario, the camera cannot recognize targets due to low light intensity but the fusion module can use the millimeter-wave radar detection result to perceive the targets. Moreover, in order to prevent a false positive alarm of the collision avoidance system, the test was conducted in an underground parking lot with a high level of clutter. Because the proposed algorithm assigns different confidences to the detected targets based on the matching results of the heterogeneous sensors, the fusion algorithm can effectively filter out clutter targets. Among the different test results, the test result is more complex in the crossroad scenario, because it maintains more complex scenario elements such as vehicles, pedestrians, and complex road structure. In terms of target tracking accuracy, the fusion algorithm optimizes the detection accuracy based on the measurement uncertainty of each sensor. Compared with the target detection and tracking results of a single sensor, the fusion algorithm achieves comprehensive optimization. This provides a decision-making foundation for assisting driving system decisions and risk management.

## 6. Conclusions

This study proposes a multimodal heterogeneous perception cross-fusion framework that combines millimeter-wave radar and camera data. It employs the Hungarian algorithm for matching and optimal estimation. To improve the estimation accuracy, the adaptive root-mean-square cubature Kalman filter is used to estimate noise characteristics and the adaptive multimodal interaction approach is introduced to improve target motion prediction. The improved joint probability data association handles multi-source perception uncertainty. Experimental results demonstrate the purposed fusion framework can enhance target tracking accuracy and robustness in complex traffic scenarios. The research has significant implications for collision avoidance systems, offering potential for more efficient fusion algorithms. Furthermore, more efficient and effective solutions for improving the fusion algorithm could be developed by considering the scenario understanding in the future study.

## Figures and Tables

**Figure 1 sensors-23-06044-f001:**
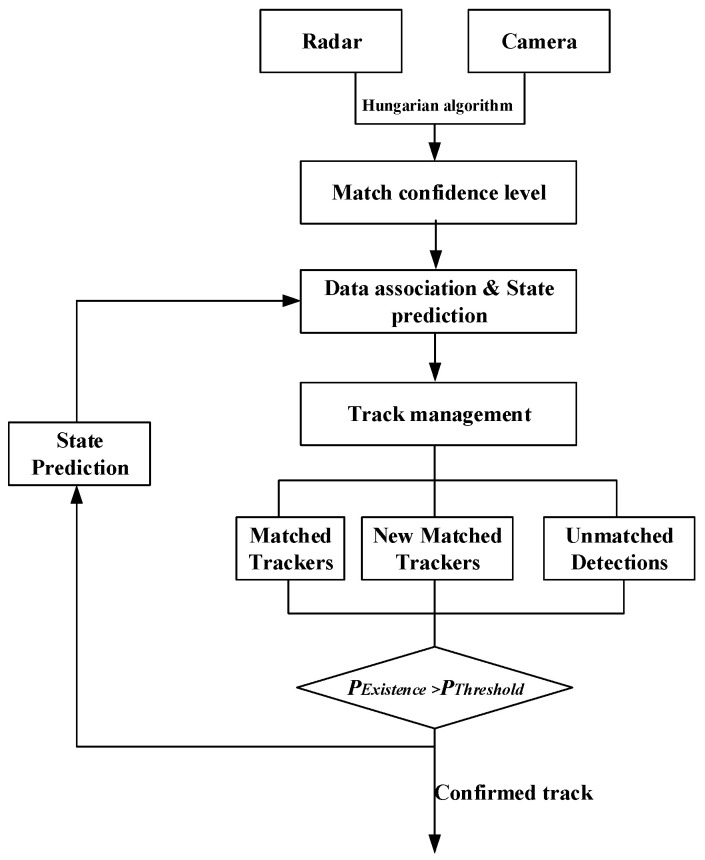
Framework for radar and camera perception information fusion.

**Figure 2 sensors-23-06044-f002:**
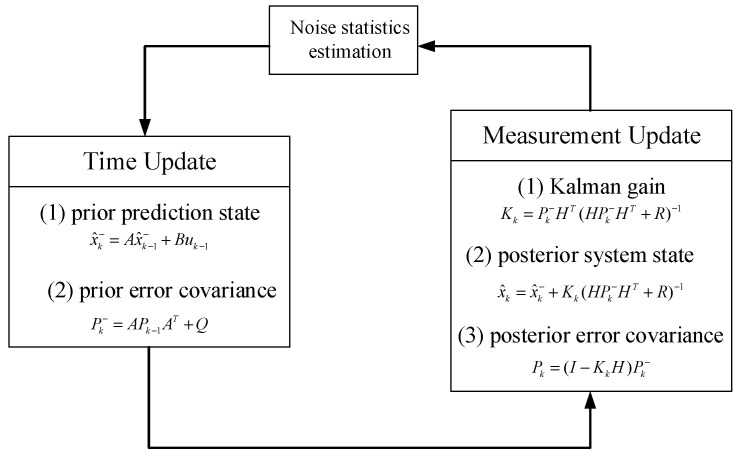
Flowchart of noise statistics estimation.

**Figure 3 sensors-23-06044-f003:**
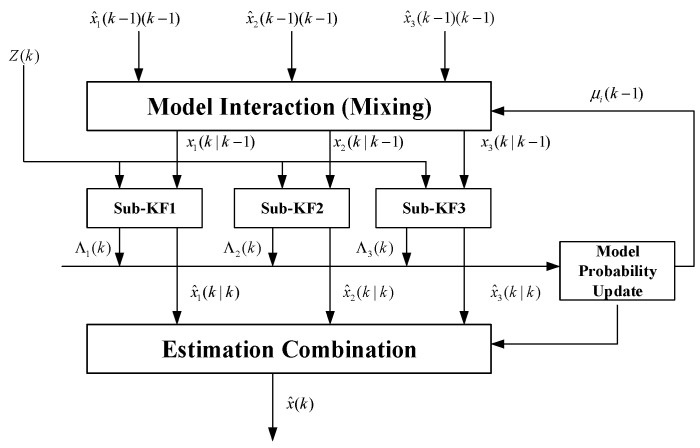
Flowchart of multiple-model interaction algorithm.

**Figure 4 sensors-23-06044-f004:**
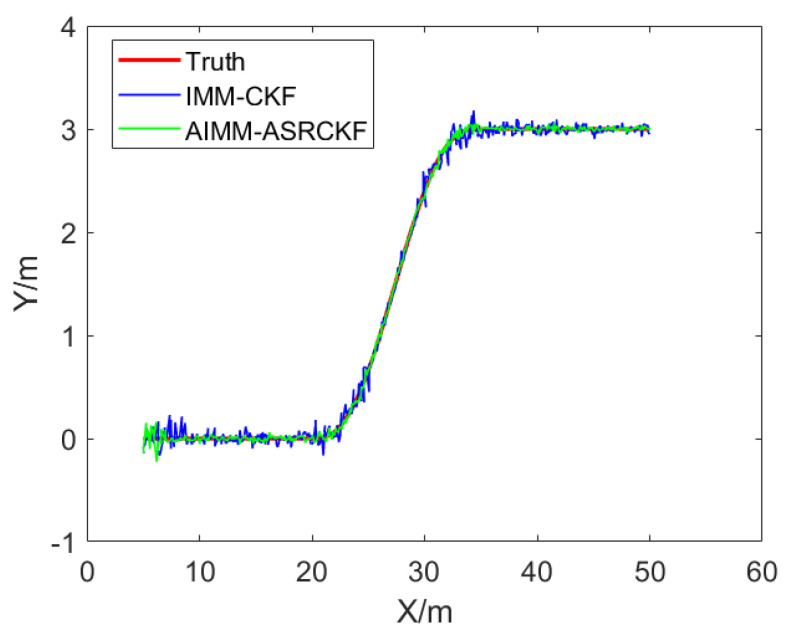
Filtering effect comparison between AIMM−ASRCKF and IMM−CKF.

**Figure 5 sensors-23-06044-f005:**
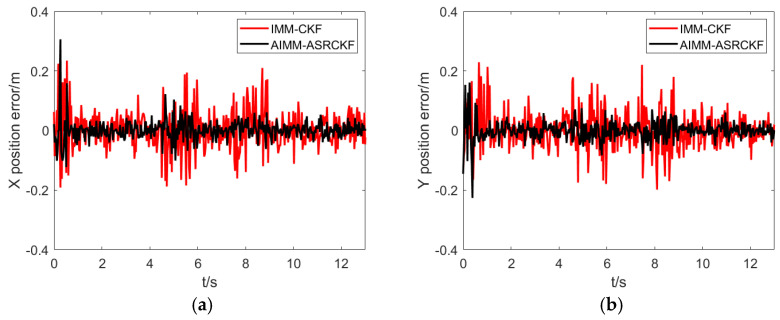
Position error comparison between different filtering algorithm. (**a**) X direction. (**b**) Y direction.

**Figure 6 sensors-23-06044-f006:**
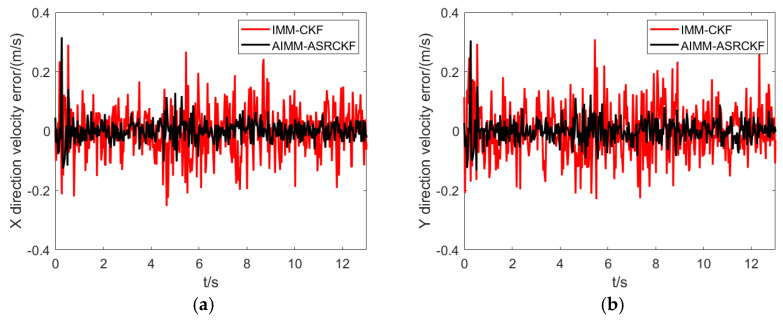
Velocity error comparison between different filtering algorithm. (**a**) X direction. (**b**) Y direction.

**Figure 7 sensors-23-06044-f007:**
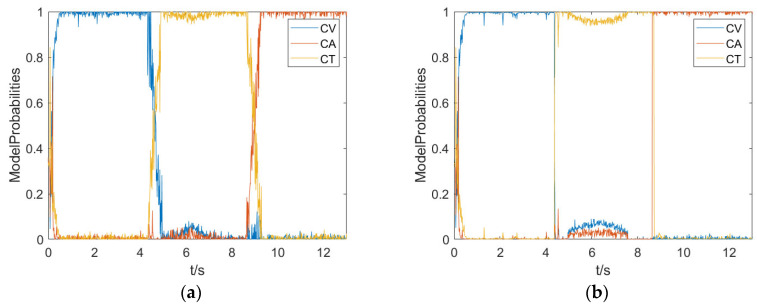
The comparison of the model probability. (**a**) IMM-CKF model probability. (**b**) AIMM-ASCRKF model probability.

**Figure 8 sensors-23-06044-f008:**
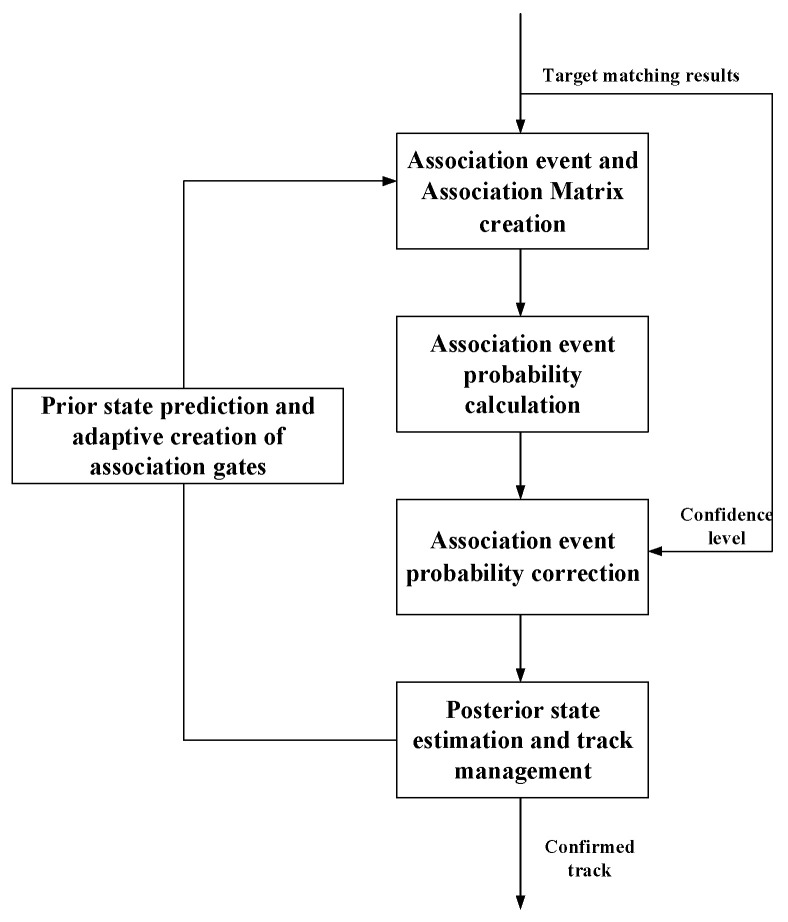
Improved JPDA algorithm flowchart.

**Figure 9 sensors-23-06044-f009:**
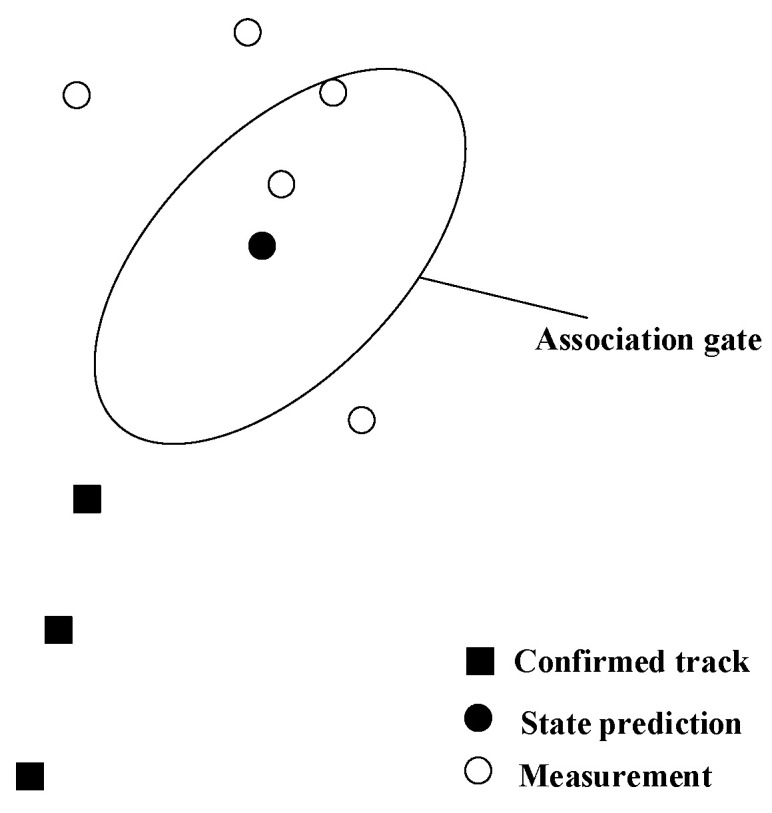
Schematic diagram of association gate.

**Figure 10 sensors-23-06044-f010:**
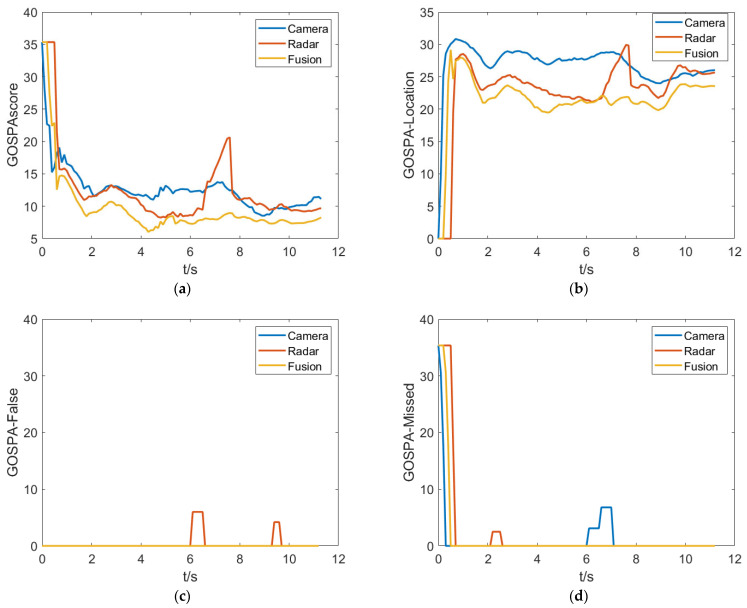
GOSPA metrics for the purposed algorithm in simulation test. (**a**) GOSPAscore. (**b**) GOSPA-Location. (**c**) GOSPA-False. (**d**) GOSPA-Missed.

**Figure 11 sensors-23-06044-f011:**
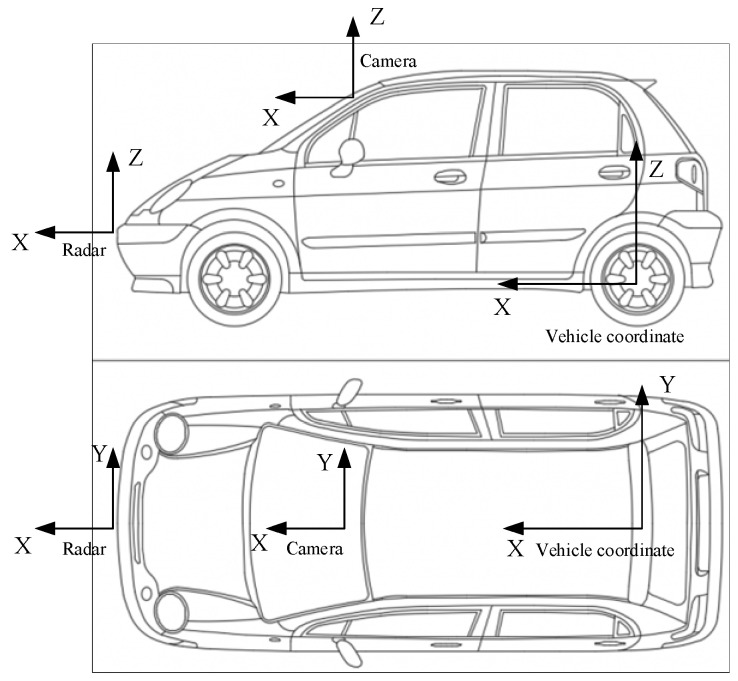
Radar, camera position, and vehicle coordinate.

**Figure 12 sensors-23-06044-f012:**
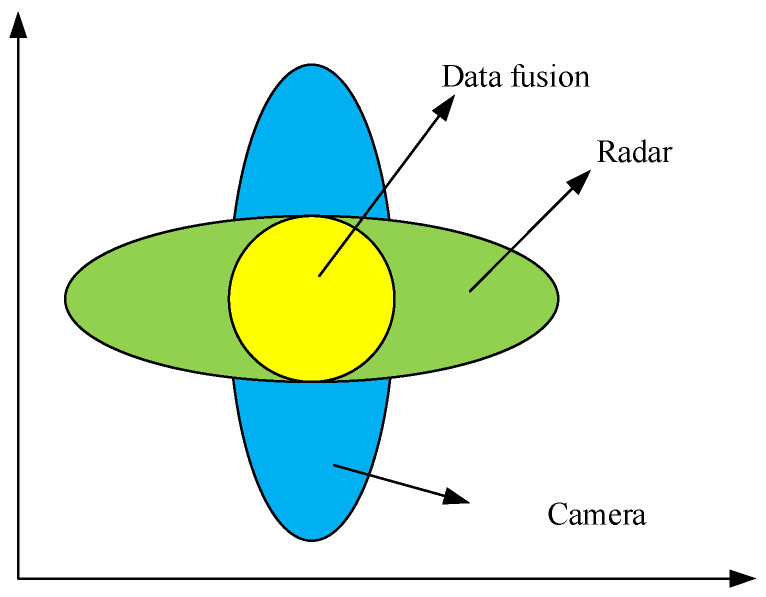
Schematic diagram of matched information fusion between radar and camera.

**Figure 13 sensors-23-06044-f013:**
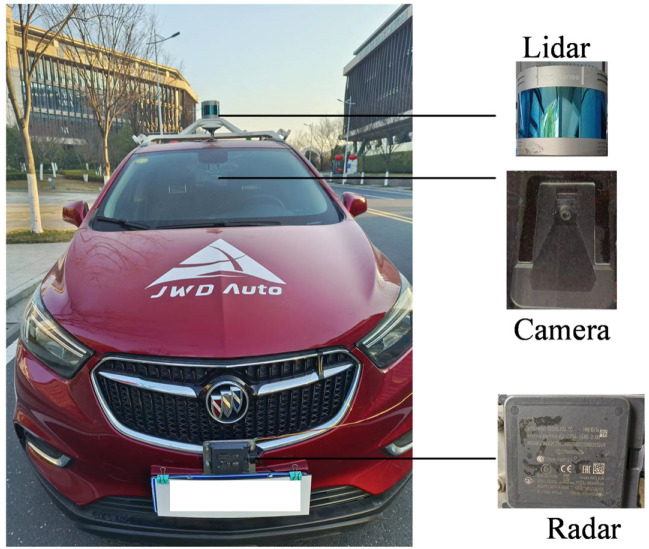
Test vehicle and sensors.

**Figure 14 sensors-23-06044-f014:**
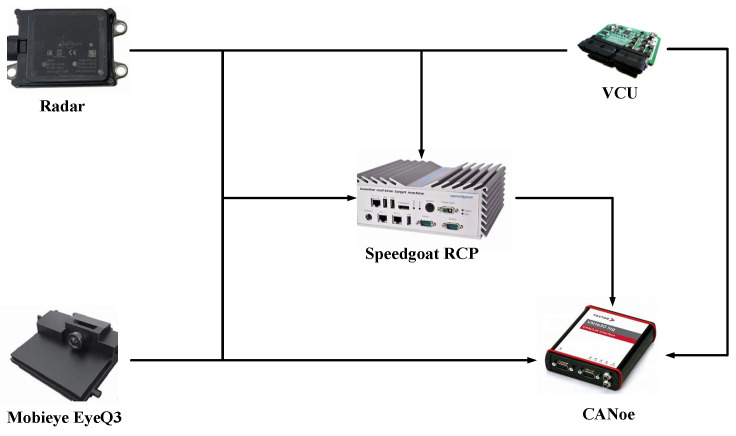
Experiment system architecture.

**Figure 15 sensors-23-06044-f015:**
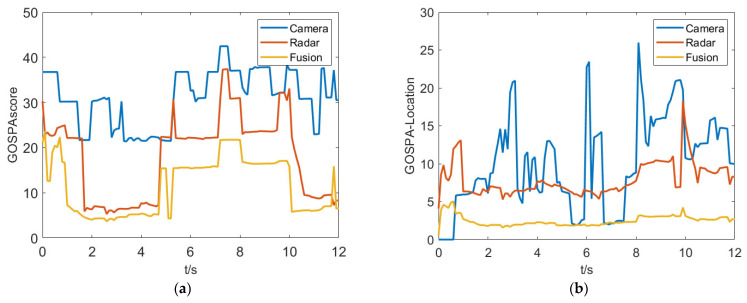
Experiment results in the crossroad scenario. (**a**) GOSPAscore. (**b**) GOSPA-Location. (**c**) GOSPA-False. (**d**) GOSPA-Missed.

**Figure 16 sensors-23-06044-f016:**
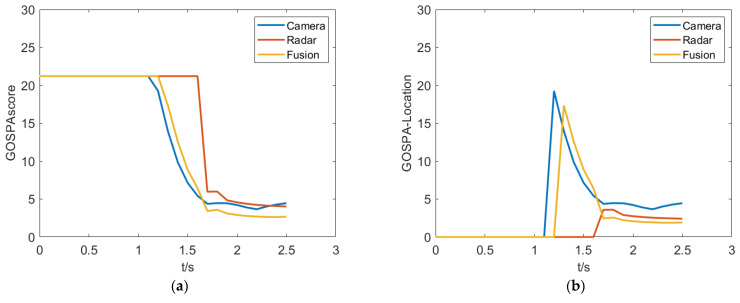
Experiment results in the pedestrian crossing scenario. (**a**) GOSPAscore. (**b**) GOSPA-Location. (**c**) GOSPA-False. (**d**) GOSPA-Missed.

**Figure 17 sensors-23-06044-f017:**
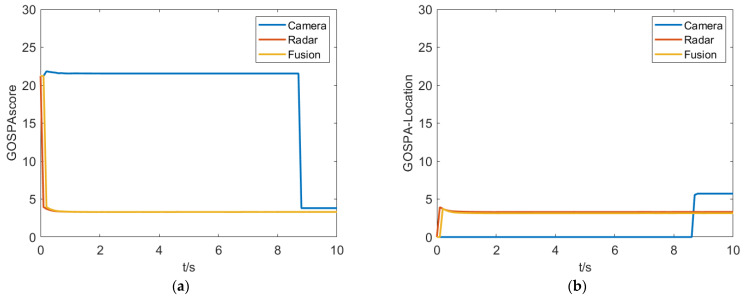
Experiment results in the nighttime scenario. (**a**) GOSPAscore. (**b**) GOSPA-Location. (**c**) GOSPA-False. (**d**) GOSPA-Missed.

**Figure 18 sensors-23-06044-f018:**
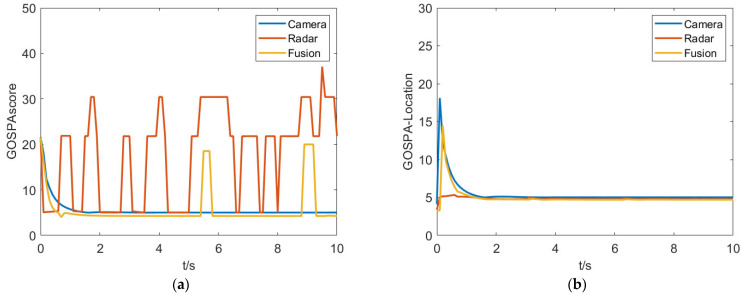
Experiment results in the underground parking scenario. (**a**) GOSPAscore. (**b**) GOSPA-Location. (**c**) GOSPA-False. (**d**) GOSPA-Missed.

**Table 1 sensors-23-06044-t001:** RMSE comparison with different filtering algorithm.

State	Direction	IMM-CKF	AIMM-ASCRKF
Position	*x*	0.084	0.021
	*y*	0.095	0.026
Velocity	*v_x_*	0.135	0.036
	*v* * _y_ *	0.127	0.078

## Data Availability

Data sharing not applicable.

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
