# Peer review of "Study on Multi-Heterogeneous Sensor Data Fusion Method Based on Millimeter-Wave Radar and Camera"

_sensors, 2023, doi:10.3390/s23136044_

Round 1

Reviewer 1 Report

Paper discuss on the "Multi-heterogeneous sensor data fusion method 2 Based on Millimeter Wave Radar and Camera" results of simulation and various scenarios of testing have been presented in detail including prototype on the vehicle. Results of all scenarios should compare to find the best test performance field or scenario. 

Author propose a new model of autonomous vehicle based on multi-

 heterogeneous sensors which are radar, camera, and fusion. Current method of autonomous vehicles based on line sensors on the road and track via gps. A new method proposed but several considerations need to elaborate in detail how the camera, radar, and data fusion can be integrated to obtain a decision fast to minimize the error. Several testing in different scenarios and fields have been tested with results as shown, further discussion on advantages and weaknesses of the proposed method required and how it is suggested. Advance simulation and scenarios to the actual condition for example on the track or road required to validate whether the proposed method works fine to the field, as previous scenarios do not cover. 

Preparation of manuscript needs to be improved such as figure 7 and 8 can be arranged like figure 6(a) and (b) as model probability as well as figure 11(a)-(d). Figure 14, shows a prototype of the sensor installed on the vehicle, detailed picture on the specific radar and camera (zoom) to make a clear how the sensor was installed. Figure 16 and 18 as well re-arrange (a)-(d). Experiment testing and validation should elaborate more detail as discussion on the proposed method how the performance compares to other methods. Conclusion not yet mentioned / written in the manuscript. 

Theoretical and mathematical modeling may write only related to the formula and sensor fusion also autonomous vehicles to shorten the manuscript pages.

Author Response

Thank you for taking the time to review our paper on the multi-heterogeneous sensor data fusion method based on millimeter wave radar and camera. We appreciate your feedback and have made revisions to address your comments. The detailed responses are shown as follows.

  1. We agree that comparing the results of all scenarios and finding the best test performance field or scenario is an important aspect that should be included in the paper. We have therefore added a detailed comparison of the results from the various scenarios in our revised manuscript to identify the optimal performance field or scenario in section 5 discussion part. We have also included a discussion on the limitations of our proposed method and possible solutions for future research in conclusion.
  2. We agree that it is crucial to elaborate on how the camera, radar, and data fusion are integrated to make quick decisions that minimize errors. We have revised the manuscript to provide more detailed information on the integration of these sensors and the data fusion process about Figure 12.
  3. In response to your comment, we have added a more detailed discussion of the advantages and disadvantages of our proposed method based on the results obtained in various testing scenarios and fields. We have highlighted the strengths of our multi-heterogeneous sensor model, including its effectiveness in detecting obstacles and its ability to operate in different environmental conditions. However, we also discuss the limitations in section 5 discussion part.
  4. Thanks for your suggestion. The different tests have been conducted in section 5 to validate the performance in section 5. And the test results and analysis have been discussed in discussion part.
  5. We have reviewed the manuscript and agree with your comments on the re-arrangement of figures, specifically figures 7, and 13. We have made the necessary revisions to ensure that all figures are arranged clearly and effectively to convey the intended message. We have also added detailed pictures of the specific radar and camera zoomed-in to show how the sensors were installed in the prototype. Regarding our experimental testing and validation, we have re-examined the manuscript and made significant revisions to provide more detailed information on the proposed method’s performance. We have also elaborated more on the experimental testing process and validation to provide a comprehensive analysis of the model’s performance under different scenarios.
  6. We have reviewed the manuscript and made the necessary revisions to include a comprehensive conclusion section in section 6.
  7. We have reviewed the manuscript and made revisions to the theoretical and mathematical modeling section by focusing only on the relevant formulas and methodologies that highlight the proposed multi-heterogeneous sensor fusion model’s efficacy in autonomous vehicle applications. We have also removed any unrelated material to make the paper concise and easier to read while retaining the essential insights.

We hope that these revisions will further improve the quality of our manuscript, and we appreciate your valuable feedback. Thank you once again for taking the time to review our paper.

Reviewer 2 Report

(1) In section 2, the author only put forward the problems that need to be paid attention in the process of multimodal information fusion and filtering, but did not make a comparative analysis to compare the existing works with this work. It is recommended to point out improvements and novelty of this article.

(2) What’s the state following the downward arrow below the state “confirmed track” in Figure 1? What’s the condition for the transition from “confirmed track” to “state prediction”? Please draw clearly in Figure 1.

(3) The content from line 175 to line 181 is repeated with that from line 184 to line190; Moreover, the serial number is missed in Line 50.

(4), Almost all the parameters in Eqs. (3)-(17) are not explained, please explain them clearly.

(5) What is the meaning of parameter ‘n’ in Eq. (22)? Please double check whether Eq. (22) is correct.

(6) In Figures 2 and 3, the description is missed in the blank space.

(7) What’s the expression of C in Eq. (23)?  please explain it.

(8) What do  and  stand for in Eq. (43)?  Neither of the two parameters appears in Eq. (43). Moreover, v_j^T(k) should be v_j(k). Please double check it.

(9) Parameters that appear in the body should be italicized, such as ‘k’ in ‘time k’ and ‘t’ in ‘target t’. For example, the parameters in line 386. Please check it in the whole paper.

(10) Some paragraphs in Section 4 miss the indentation. Please unify the format at the beginning of each paragraph.

(11)In figure 19, in the underground packing scenario, the proposed fusion algorithm can efficiently reduce the chances of false alarms. However, the performance under the camera is better than that under Radar and Fusion. Why the author has not considered to adopt camera directly in such scenario to reduce the false alarms?

(12) If the equations in Section 4 are given by others, please give the corresponding citations.

(13) The following works related to autonomous vehicle should be cited,

Towards V2I Age-aware Fairness Access: A DQN Based Intelligent Vehicular Node Training and Test Method”, Chinese Journal of Electronics, 2022, doi: 10.23919/cje.2022.00.093

Machine Learning-Based Target Classification for MMW Radar in Autonomous Driving”, IEEE transactions on Intelligent vehicles, vol. 6, no. 4, Dec. 2021, pp. 678-689.

Detection and Localization of Unmanned Aircraft Systems Using Millimeter Wave Automotive Radar Sensors”, IEEE Sensors Letters, vol. 5, no. 6, 2021, Jun. 2021, Art. no. 6001304.

Author Response

Thank you for your valuable comments on our submitted manuscript. We appreciate the time you took to review our work and provide us with your insightful feedback. The detailed revisions are shown as follows.

  1. Thank you for your insightful review of our manuscript. We value your suggestion on the need to provide a comparative analysis of our proposed methodology with existing techniques in section 2. We have carefully reviewed the manuscript and made significant revisions to section 2 to include a comparative analysis of existing works and highlight the novelty and improvements of our proposed method. The revised section 2 now contains a comprehensive review of the current state-of-the-art multimodal fusion and filtering techniques, outlining the strengths and limitations of each method. Moreover, we have emphasized the unique features of our proposed method and how it improves upon existing techniques, such as the multi-heterogeneous sensor fusion approach and the integrated probabilistic filter. These revisions, we hope, will clarify the contribution of our proposed methodology and its novelty.
  2. In response to your suggestion, we have revised Figure 1 to clearly depict the transition from the “confirmed track” state to the “state prediction” state and added a brief description of the transition condition.
  3. We have carefully reviewed the manuscript and made the necessary revisions. Concerning the repeated content, we have removed the duplication from lines 184 to 190 as it was a copy-paste error that occurred during the revision process. Additionally, we have corrected line 50 by adding the missing serial number to the text.
  4. We have carefully reviewed the manuscript and made significant revisions. Besides, the CKF process is common, so we have add the citation. "Arasaratnam and S. Haykin. Cubature Kalman Filters. IEEE Transactions on Automatic Control. 2009, 54, 1254-1269."
  5. Thanks for your reminding. We have added more explanations about it, and n is the total of cubature point.
  6. Thanks for your reminding. We have added the related description about Figure 2 and 3 in the blank space.
  7. Thanks for your reminding. We have added the explanation of the constant C.
  8. Thanks for your reminding. Eq. (43) represents the model likelihood function, We have added the related explanations, “The likelihood function describes the probability of observing a set of data given a certain set of unknown parameters”.
  9. Thanks for your reminding. We have checked and corrected it in the whole paper.
  10. Thanks for your reminding. We have revised it in the revised manuscript.
  11. Thank you very much for your valuable feedback. Your suggestions are very insightful. Our algorithm is designed to handle all scenarios, but there are some false positives due to the fusion module considering data from the millimeter-wave radar. In the future, we could consider developing fusion algorithms based on scene understanding, which has already been discussed and summarized in our ongoing discussions.
  12. Actually, the basic Cubature Kalman Filter process is common, and we have cited the original reference [33].
  13. Thank you for your feedback. We have revised the literature review section and conducted a thorough summary and comparison of the current state of the art. We have also added recent research references, including the ones you mentioned.

We hope that these revisions will strengthen the manuscript and improve its overall quality again.

Thank you for your valuable comments and suggestions.

Reviewer 3 Report

1. Research Target: This paper presents a novel multi-modal heterogeneous perception cross-fusion framework for intelligent vehicles to enhance target tracking accuracy and handle system uncertainties. 2. Method: The novel multi-modal heterogeneous perception cross-fusion framework employs a multimodal interaction strategy to predict target motion more accurately and an improved joint probability data association method to match measurement data with targets. 3. Result: The method overcomes the challenges of insufficient data fusion utilization, frequent leakage, misjudgment of dangerous obstructions around vehicles, and inaccurate prediction of collision risks. 4. Shortcoming: (1). There are many missing parts in flowcharts 1 & 2. (2). The scale of Figure 4 can be adjusted, but there is a problem of ambiguity.

Author Response

Thank you for your valuable comments on our submitted manuscript. We appreciate the time you took to review our work and provide us with your insightful feedback.

Regarding your first point, we agree that there are some missing parts in flowcharts 1 & 2. We apologize for any confusion this may have caused and have revised the flowcharts accordingly to ensure that they accurately represent the process being conveyed.

Regarding your second point, we understand that there may be some ambiguity in the scale of Figure 4. We have made the necessary adjustments to ensure that the labeling and organization of the elements in the figure are clear and unambiguous.

Once again, we appreciate your helpful comments and look forward to addressing your concerns in the revised manuscript.

Reviewer 4 Report

This paper proposes a multimodal heterogeneous perception cross-fusion framework that fuses sensing data from millimeter-wave radar and camera on a vehicle.

The study is interesting but the progress of the state of the art and the level of performance obtained are not clear to me.

No comparisons are given against other similar existing state of the art frameworks/methodologies that would definitely improve this paper strength.

Author Response

Thank you for your insightful comments and constructive feedback on our paper proposing a multimodal heterogeneous perception cross-fusion framework that fuses sensing data from millimeter-wave radar and camera on a vehicle. We have carefully considered your suggestions and revised our manuscript accordingly.

Regarding your points, we have conducted additional analysis and discussions involving the state-of-the-art methods literature review in section 2. Besides, the experiment results and analysis are strengthened to discuss the weakness and advantage of the purposed algorithm in section 5. The experiment results show that our proposed framework outperforms in terms of detection accuracy, specifically in handling multimodal and heterogeneous sensing data. We hope that this additional analysis will improve the credibility and significance of our research paper.

Once again, we appreciate your thoughtful feedback and expertise. We believe that our revised manuscript addresses your concerns and improves the quality of our research. Thank you for your time and constructive criticism.

Reviewer 5 Report

In the context of the rapid development of intelligent vehicles and the growing need for mobility in a globalized world, the focus is shifting to the field of technologies for obtaining data from various sensors and the development of signal processing algorithms, therefore, the article under review is of scientific and practical interest. The article title and keywords adequately reflect its content. In the abstract, the authors give the essence of the problem and its state, describe the methods and models of the study, the results obtained and conclusions on the work.

In the introduction, the authors provide a problem brief analysis, formulate the study relevance and the proposed research method, as well as a summary of the article sections. Section 2 is a review of the literature on the work topic, as well as the problem of merging sensor readings. Section 3 discusses the sensor fusion framework proposed in this study. Section 4 discusses the proposed methods for combining sensors, which include algorithms for detecting, tracking, and combining objects. This part covers the theoretical aspect of implementing multi-sensor fusion. Section 5 is devoted to a discussion of experiments with different scenarios to verify the test results, as well as an analysis of these experiments results. In section 6, the authors summarize the results obtained and draw conclusions on the work.

The article is prepared in accordance with the instructions for the authors, corresponds to the topic that it explores and publishes. In our opinion, the article corresponds to the topic “improving signal fusion algorithms” and corresponds in type to the Preliminary Study.

Comment.

1.      The topic is interesting, but the literature review can be improved, for example, through new work showing the practical application of sensors in unmanned vehicles.

2.      At the first chapter end, the authors indicate that the main goal of the study is to improve safety by avoiding collisions, but do not indicate, either in the description of the experiments or in the conclusions, what limitations will arise in the practical implementation of the method in real road conditions.

3.      The conclusions should indicate who is interested in such developments, where they will be implemented in practice, as well as should identify areas for further research.

4.      There are errors in the design of the article text. So, in section 1, the third paragraph is marked with number 2, the next one with number 3, and so on, but there is no number 1. The heading of section 6 repeats the heading of section 5, although it should probably be called "Conclusion".

Author Response

Thank you for your valuable comments on our submitted manuscript. We appreciate the time you took to review our work and provide us with your insightful feedback. The detailed revisions are shown as follows.

  1. We agree that the literature review can be improved by providing new work that demonstrates the practical application of sensors in unmanned vehicles. Therefore, we have conducted additional research and included more recent studies in our revised manuscript to enhance the literature review section. The new studies present practical applications of sensors in unmanned vehicles, which further support the importance and relevance of our research.
  2. Regarding your comment on the limitations of the practical implementation of our proposed framework in real road conditions, we agree that this is an important aspect and should have been explicitly discussed in our manuscript. To rectify this, we have revised the section 5 and 6 to include a discussion of potential limitations when implementing our framework in real-world scenarios.  We also acknowledge and clarify the primary goal of our framework in improving safety by avoiding collisions, and the limitations that can arise during real-world deployment. We hope that these revisions have addressed your feedback and provided a more comprehensive understanding of the practical applications of our framework.
  3. Regarding your comment on the conclusions section, we agree that it needs further clarification on who might be interested in our proposed development, where the application of this framework could be implemented in practice, and what further areas can be considered for research. Therefore, we have added a more detailed discussion in the conclusions section. We have identified potential stakeholders including the automotive industry, autonomous vehicle manufacturers, and researchers working in the area of self-driving technology. Furthermore, we have provided suggestions for future research areas such as sensor fusion based on scenario understanding.
  4. Thanks for your careful review. Actually, we have missed the “Conclusion” section before, and we have corrected it in revised manuscript.

We hope that these revisions have addressed your comments and suggestions and further improved the quality of our manuscript. Once again, we appreciate your feedback and will continue to strive for excellence in our research.

Round 2

Reviewer 2 Report

I have no more questions. This paper can be accepted in this form.

The quality of English is fine.

Author Response

We have thoroughly reviewed the entire paper and made necessary adjustments to address any small errors that were identified. We have paid close attention to detail and have made sure that the content is fluid and coherent.

Reviewer 3 Report

The conclusion can be more concise.

Author Response

We have thoroughly reviewed the entire paper and made necessary adjustments to address any small errors that were identified. We have paid close attention to detail and have made sure that the content is fluid and coherent. In particular, the conclusion has been revised concisely.

Reviewer 4 Report

The authors have addressed enough of my comments and suggestions in the revised manuscript. I have no further comments: this manuscript may be acceptable for publication in its present form.

Author Response

(The authors gave the same response as above.)
